# BAFFLE: A BASELINE OF BACKPROPAGATION-FREE FEDERATED LEARNING

## ABSTRACT

Federated learning (FL) is a general principle for decentralized clients to train a server model collectively without sharing local data. FL is a promising framework with practical applications, but its standard training paradigm requires the clients to backpropagate through the model to compute gradients. Since these clients are typically edge devices and not fully trusted, executing backpropagation on them incurs computational and storage overhead as well as white-box vulnerability. In light of this, we develop backpropagation-free federated learning, dubbed BAFFLE, in which backpropagation is replaced by multiple forward processes to estimate gradients. BAFFLE is 1) memory-efficient and easily fits uploading bandwidth; 2) compatible with inference-only hardware optimization and model quantization or pruning; and 3) well-suited to trusted execution environments, because the clients in BAFFLE only execute forward propagation and return a set of scalars to the server. Empirically we use BAFFLE to train deep models from scratch or to finetune pretrained models, achieving acceptable results.

## 1 INTRODUCTION

Federated learning (FL) allows decentralized clients to collaboratively train a server model (McMahan et al., 2017). In each training round, the selected clients compute model gradients or updates on their local private datasets, without explicitly exchanging sample points to the server. While FL describes a promising blueprint and has several applications (Yang et al., 2018; Hard et al., 2018; Li et al., 2020b), the mainstream training paradigm of FL is still gradient-based that requires the clients to locally execute backpropagation, which leads to two practical limitations:

**(i) Overhead for edge devices.** The clients in FL are usually edge devices, such as mobile phones and IoT sensors, whose hardware is primarily optimized for inference-only purposes (Sharma et al., 2018; Umuroglu et al., 2018), rather than for backpropagation. Due to the limited resources, computationally affordable models running on edge devices are typically quantized and pruned (Wang et al., 2019a), making exact backpropagation difficult. In addition, standard implementations of backpropagation rely on either forward-mode or reverse-mode auto-differentiation in contemporary machine learning packages (Bradbury et al., 2018; Paszke et al., 2019b), which increases storage requirements.

**(ii) White-box vulnerability.** To facilitate gradient computing, the server regularly distributes its model status to the clients, but this white-box exposure of the model renders the server vulnerable to, e.g., poisoning or inversion attacks from malicious clients (Shokri et al., 2017; Xie et al., 2020; Zhang et al., 2020; Geiping et al., 2020). With that, recent attempts are made to exploit trusted execution environments (TEEs) in FL, which can isolate the model status within a black-box secure area and significantly reduce the success rate of malicious evasion (Chen et al., 2020; Mo et al., 2021; Mondal et al., 2021). However, TEEs are highly memory-constrained (Truong et al., 2021), while backpropagation is memory-consuming to restore intermediate states.

While numerous solutions have been proposed to alleviate these limitations (related work discussed in Appendix B), we raise an essential question: *how to perform backpropagation-free FL?* Inspired by the literature on zero-order optimization (Stein, 1981), we intend to substitute backpropagation with multiple forward or inference processes to estimate the gradients. Technically speaking, we propose the framework of **BA**ckpropagation-**F**ree **F**ederated **LE**arning (**BAFFLE**). As illustrated in Figure 1, BAFFLE consists of three conceptual steps: (1) each client locally perturbs the model parameters $2K$ times as $\mathbf{W} \pm \boldsymbol{\delta}_k$ (the server sends the random seed to clients for generating $\{\boldsymbol{\delta}_k\}_{k=1}^K$); (2) each client executes forward processes on the perturbed models using its private dataset $\mathbb{D}_c$ and obtains $K$ loss differences $\{\Delta\mathcal{L}(\mathbf{W}, \boldsymbol{\delta}_k; \mathbb{D}_c)\}_{k=1}^K$; (3) the server aggregates loss differences to estimate gradients.

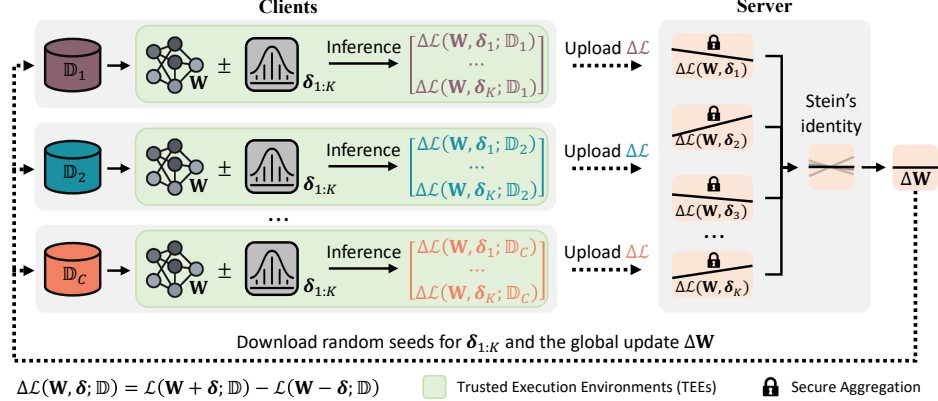

$$\Delta\mathcal{L}(\mathbf{W},\boldsymbol{\delta};\mathbb{D})=\mathcal{L}(\mathbf{W}+\boldsymbol{\delta};\mathbb{D})-\mathcal{L}(\mathbf{W}-\boldsymbol{\delta};\mathbb{D})$$  Trusted Execution Environments (TEEs)  Secure Aggregation

Figure 1: A sketch map of BAFFLE. In addition to the global parameters update $\Delta\mathbf{W}$, each client downloads random seeds to locally generate perturbations $\pm\boldsymbol{\delta}_{1:K}$ and perform $2K$ times of forward propagation (i.e., inference) to compute loss differences. The server can recover these perturbations using the same random seeds and obtain $\Delta\mathcal{L}(\mathbf{W},\boldsymbol{\delta}_k)$ by secure aggregation. Each loss difference $\Delta\mathcal{L}(\mathbf{W},\boldsymbol{\delta}_k;\mathbb{D}_c)$ is a floating-point number, so $K$ can be easily adjusted to fit the uploading bandwidth.

BAFFLE's defining characteristic is that it only utilizes forward propagation, which is memory-efficient and does not require auto-differentiation. It is well-adapted to model quantization and pruning as well as inference-only hardware optimization on edge devices. Compared to backpropagation, the computation graph of BAFFLE is more easily optimized, such as by slicing it into per-layer calculation (Kim et al., 2020). Since each loss difference $\Delta\mathcal{L}(\mathbf{W},\boldsymbol{\delta}_k;\mathbb{D}_c)$ is a scalar, BAFFLE can easily accommodate the uploading bandwidth of clients by adjusting the value of $K$ as opposed to using, e.g., gradient compression (Suresh et al., 2017). BAFFLE is also compatible with recent advances in inference approaches for TEE (Tramer & Boneh, 2019; Truong et al., 2021), providing an efficient solution for combining TEE into FL and preventing white-box evasion.

We adapt secure aggregation (Bonawitz et al., 2017a) to zero-order optimization and investigate ways to improve gradient estimation in BAFFLE. Empirically, BAFFLE is used to train models from scratch on MNIST (LeCun et al., 1998) and CIFAR-10/100 (Krizhevsky & Hinton, 2009), and transfer ImageNet-pretrained models to OfficeHome (Venkateswara et al., 2017). Compared to conventional FL, BAFFLE achieves suboptimal but acceptable performance. These results shed light on the potential of BAFFLE and general backpropagation-free methods in FL.

## 2 PRELIMINARIES

**Finite difference.** Gradient-based optimization techniques (either first-order or higher-order) are the most frequently used tools to train deep networks (Goodfellow et al., 2016). Nevertheless, recent progress demonstrates promising applications of zero-order optimization methods for training, particularly when exact derivatives cannot be obtained (Flaxman et al., 2004; Nesterov & Spokoiny, 2017; Liu et al., 2020a) or backward processes are computationally prohibitive (Pang et al., 2020; He et al., 2022). Zero-order approaches require only multiple forward processes that may be executed in parallel. Along this routine, finite difference stems from the definition of derivatives and can be generalized to higher-order and multivariate cases by Taylor's expansion. For any differentiable loss $\mathcal{L}(\mathbf{W};\mathbb{D})$ and a small perturbation $\boldsymbol{\delta}\in\mathbb{R}^n$, finite difference employs the *forward difference scheme*

$$\mathcal{L}(\mathbf{W}+\boldsymbol{\delta};\mathbb{D})-\mathcal{L}(\mathbf{W};\mathbb{D})=\boldsymbol{\delta}^\top\nabla_\mathbf{W}\mathcal{L}(\mathbf{W};\mathbb{D})+o(\|\boldsymbol{\delta}\|_2), \tag{1}$$

where $\boldsymbol{\delta}^\top\nabla_\mathbf{W}\mathcal{L}(\mathbf{W};\mathbb{D})$ is a scaled directional derivative along $\boldsymbol{\delta}$. Furthermore, we can use the *central difference scheme* to obtain higher-order residuals as

$$\mathcal{L}(\mathbf{W}+\boldsymbol{\delta};\mathbb{D})-\mathcal{L}(\mathbf{W}-\boldsymbol{\delta};\mathbb{D})=2\boldsymbol{\delta}^\top\nabla_\mathbf{W}\mathcal{L}(\mathbf{W};\mathbb{D})+o(\|\boldsymbol{\delta}\|_2^2). \tag{2}$$

**Federated learning.** Suppose we have $C$ clients, and the $c$-th client's private dataset is defined as $\mathbb{D}_c:=\{(\mathbf{X}_i^c,\mathbf{y}_i^c)\}_{i=1}^{N_c}$ with $N_c$ input-label pairs. Let $\mathcal{L}(\mathbf{W};\mathbb{D}_c)$ represent the loss function for the dataset $\mathbb{D}_c$, where $\mathbf{W}\in\mathbb{R}^n$ denotes the server model's global parameters. The training objective of FL is to find $\mathbf{W}$ that minimize the total loss function as

$$\mathcal{L}(\mathbf{W}):=\sum_{c=1}^C\frac{N_c}{N}\mathcal{L}(\mathbf{W};\mathbb{D}_c),\text{ where }N=\sum_{c=1}^C N_c. \tag{3}$$

---

**Algorithm 1** Backpropagation-free federated learning (BAFFLE)

---

1: **Notations:** Se denotes the operations done on servers; Cl denotes the operations done on clients; TEE for the TEE module; and $\Rightarrow$ denotes the communication process.

2: **Inputs:** $C$ clients with local dataset $\{\mathbb{D}_c\}_{c=1}^C$ containing $N_c$ input-label pairs, $N = \sum_{c=1}^C N_c$; learning rate $\eta$, training iterations $T$, perturbation number $K$, noise scale $\sigma$.

3: Se: initializing model parameters $\mathbf{W} \leftarrow \mathbf{W}_0$;

4: Se: encoding the computing paradigm into TEE as $\text{TEE} \circ \Delta\mathcal{L}(\mathbf{W}, \boldsymbol{\delta}; \mathbb{D})$;          # *optional*

5: **for** $t = 0$ **to** $T-1$ **do**

6:     Se $\Rightarrow$ all Cl: downloading model parameters $\mathbf{W}_t$ and the computing paradigm;

7:     Se $\Rightarrow$ all Cl: downloading the random seed $s_t$;          # *4 Bytes*

8:     Se: sampling $K$ perturbations $\{\boldsymbol{\delta}_k\}_{k=1}^K$ from $\mathcal{N}(0, \sigma^2\mathbf{I})$ using the random seed $s_t$;

9:     all Cl: negotiating a group of zero-sum noises $\{\boldsymbol{\epsilon}_c\}_{c=1}^C$ for secure aggregation;

10:     **for** $c = 1$ **to** $C$ **do**

11:         Cl: sampling $K$ perturbations $\{\boldsymbol{\delta}_k\}_{k=1}^K$ from $\mathcal{N}(0, \sigma^2\mathbf{I})$ using the random seed $s_t$;

12:         Cl: computing $\text{TEE} \circ \Delta\mathcal{L}(\mathbf{W}_t, \boldsymbol{\delta}_k; \mathbb{D}_c)$ via forward propagation for each $k$;

13:         Cl $\Rightarrow$ Se: uploading $K$ outputs $\left\{ \text{TEE} \circ \Delta\mathcal{L}(\mathbf{W}_t, \boldsymbol{\delta}_k; \mathbb{D}_c) + \frac{N}{N_c}\boldsymbol{\epsilon}_c \right\}_{k=1}^K$;     # *$4 \times K$ Bytes*

14:     **end for**

15:     Se: aggregating $\Delta\mathcal{L}(\mathbf{W}_t, \boldsymbol{\delta}_k) \leftarrow \sum_{c=1}^C \frac{N_c}{N} \left[ \text{TEE} \circ \Delta\mathcal{L}(\mathbf{W}_t, \boldsymbol{\delta}_k; \mathbb{D}_c) + \frac{N}{N_c}\boldsymbol{\epsilon}_c \right]$ for each $k$;

16:     Se: computing $\widehat{\nabla_{\mathbf{W}_t}}\mathcal{L}(\mathbf{W}_t) \leftarrow \frac{1}{K}\sum_{k=1}^K \frac{\boldsymbol{\delta}_k}{2\sigma^2}\Delta\mathcal{L}(\mathbf{W}_t, \boldsymbol{\delta}_k)$;

17:     Se: $\mathbf{W}_{t+1} \leftarrow \mathbf{W}_t - \eta\widehat{\nabla_{\mathbf{W}_t}}\mathcal{L}(\mathbf{W}_t)$;

18: **end for**

19: **Return:** final model parameters $\mathbf{W}_T$.

---

In the conventional FL framework, clients locally compute gradients $\{\nabla_{\mathbf{W}}\mathcal{L}(\mathbf{W}; \mathbb{D}_c)\}_{c=1}^C$ or model updates through backpropagation and then upload them to the server. Federated average (McMahan et al., 2017) performs global aggregation using $\Delta\mathbf{W} := \sum_{i=1}^C \frac{N_c}{N}\Delta\mathbf{W}_c$, where $\Delta\mathbf{W}_c$ is the local update obtained via executing $\mathbf{W}_c \leftarrow \mathbf{W}_c - \eta\nabla_{\mathbf{W}_c}\mathcal{L}(\mathbf{W}_c; \mathbb{D}_c)$ multiple times and $\eta$ is learning rate.

**Zeroth-order FL.** Similar to our work, DLZO (Li & Chen, 2021) and FedZO (Fang et al., 2022) present zeroth-order optimization methods for FL independently in batch-level and epoch-level communications. However, they concentrate primarily on basic linear models with softmax regression problems and ignore deep models. Besides, they also do not account for server security aggregation in conjunction with zero-order optimization. In comparison, BAFFLE enables security aggregation, can train deep models such as WideResNet from scratch and achieves reasonable results, e.g. 95.17% accuracy on MNIST with 20 communication rounds versus 83.58% for FedZO with 1,000 rounds.

## 3 BACKPROPAGATION-FREE FEDERATED LEARNING

In this section, we introduce zero-order optimization into FL and develop BAFFLE, a backpropagation-free federated learning framework that uses multiple forward processes in place of backpropagation. An initial attempt is to apply finite difference as the gradient estimator. To estimate the full gradients, we need to perturb each parameter $w \in \mathbf{W}$ once to approximate the partial derivative $\frac{\partial\mathcal{L}(\mathbf{W}; \mathbb{D})}{\partial w}$, causing the forward computations to grow with $n$ (recall that $\mathbf{W} \in \mathbb{R}^n$) and making it difficult to scale to large models. In light of this, we resort to Stein's identity (Stein, 1981) to obtain an unbiased estimation of gradients from loss differences calculated on various perturbations. As depicted in Figure 1, BAFFLE clients need only download random seeds and global parameters update, generate perturbations locally, execute multiple forward propagations and upload loss differences back to the server. Furthermore, we also present convergence analyses of BAFFLE, which provides guidelines for model design and acceleration of training.

### 3.1 UNBIASED GRADIENT ESTIMATION WITH STEIN'S IDENTITY

Previous work on sign-based optimization (Moulay et al., 2019) demonstrates that deep networks can be effectively trained if the majority of gradients have proper signs. Thus, we propose performing forward propagation multiple times on perturbed parameters, in order to obtain a stochastic estimation of gradients without backpropagation. Specifically, assuming that the loss function $\mathcal{L}(\mathbf{W}; \mathbb{D})$ is

continuously differentiable w.r.t. $\mathbf{W}$ given any dataset $\mathbb{D}$, which is true (almost everywhere) for deep networks using non-linear activation functions, we define a smoothed loss function as:

$$\mathcal{L}_\sigma(\mathbf{W}; \mathbb{D}) := \mathbb{E}_{\boldsymbol{\delta} \sim \mathcal{N}(0, \sigma^2 \mathbf{I})} \mathcal{L}(\mathbf{W} + \boldsymbol{\delta}; \mathbb{D}), \tag{4}$$

where the perturbation $\boldsymbol{\delta}$ follows a Gaussian distribution with zero mean and covariance $\sigma^2 \mathbf{I}$. Stein (1981) proves the *Stein's identity* as (we recap the proof in Appendix A)

$$\nabla_{\mathbf{W}} \mathcal{L}_\sigma(\mathbf{W}; \mathbb{D}) = \mathbb{E}_{\boldsymbol{\delta} \sim \mathcal{N}(0, \sigma^2 \mathbf{I})} \left[ \frac{\boldsymbol{\delta}}{2\sigma^2} \Delta \mathcal{L}(\mathbf{W}, \boldsymbol{\delta}; \mathbb{D}) \right], \tag{5}$$

where $\Delta \mathcal{L}(\mathbf{W}, \boldsymbol{\delta}; \mathbb{D}) := \mathcal{L}(\mathbf{W} + \boldsymbol{\delta}; \mathbb{D}) - \mathcal{L}(\mathbf{W} - \boldsymbol{\delta}; \mathbb{D})$ is the loss difference. Note that computing a loss difference only requires the execution of two forwards $\mathcal{L}(\mathbf{W} + \boldsymbol{\delta}; \mathbb{D})$ and $\mathcal{L}(\mathbf{W} - \boldsymbol{\delta}; \mathbb{D})$ without backpropagation. It is trivial that $\mathcal{L}_\sigma(\mathbf{W}; \mathbb{D})$ is continuously differentiable for any $\sigma \geq 0$ and $\nabla_{\mathbf{W}} \mathcal{L}_\sigma(\mathbf{W}; \mathbb{D})$ converges uniformly as $\sigma \to 0$; hence, it follows that $\nabla_{\mathbf{W}} \mathcal{L}(\mathbf{W}; \mathbb{D}) = \lim_{\sigma \to 0} \nabla_{\mathbf{W}} \mathcal{L}_\sigma(\mathbf{W}; \mathbb{D})$. Therefore, we can obtain a stochastic estimation of gradients using Monte Carlo by 1) selecting a small value of $\sigma$; 2) randomly sampling $K$ perturbations from $\mathcal{N}(0, \sigma^2 \mathbf{I})$ as $\{\boldsymbol{\delta}_k\}_{k=1}^K$; and 3) utilizing the Stein's identity in Eq. (5) to calculate

$$\widehat{\nabla_{\mathbf{W}}} \mathcal{L}(\mathbf{W}; \mathbb{D}) := \frac{1}{K} \sum_{k=1}^K \left[ \frac{\boldsymbol{\delta}_k}{2\sigma^2} \Delta \mathcal{L}(\mathbf{W}, \boldsymbol{\delta}_k; \mathbb{D}) \right]. \tag{6}$$

## 3.2 Operating flow of BAFFLE

Based on the forward-only gradient estimator $\widehat{\nabla_{\mathbf{W}}} \mathcal{L}(\mathbf{W}; \mathbb{D})$ derived in Eq. (6), we outline the basic operating flow of our BAFFLE system as in Algorithm 1 as follows:

**Model initialization.** (Lines 3~4, done by server) The server initializes the model parameters to $\mathbf{W}_0$ and optionally encodes the computing paradigm of loss differences $\Delta \mathcal{L}(\mathbf{W}, \boldsymbol{\delta}; \mathbb{D})$ into the TEE module (see Appendix C for more information on TEE);

**Downloading paradigms.** (Lines 6~7, server $\Rightarrow$ all clients) In round $t$, the server distributes the most recent model parameters $\mathbf{W}_t$ (or the model update $\Delta \mathbf{W}_t$) and the computing paradigm to all the $C$ clients. In addition, in BAFFLE, the server sends a random seed $s_t$ (rather than directly sending the perturbations to reduce communication burden);

**Local computation.** (Lines 11~12, done by clients) Each client generates $K$ perturbations $\{\boldsymbol{\delta}_k\}_{k=1}^K$ locally from $\mathcal{N}(0, \sigma^2 \mathbf{I})$ using random seed $s_t$, and executes the computing paradigm to obtain loss differences. $K$ is chosen adaptively based on clients' computation capability;

**Uploading loss differences.** (Line 13, all clients $\Rightarrow$ server) Each client uploads $K$ noisy outputs $\{\Delta \mathcal{L}(\mathbf{W}_t, \boldsymbol{\delta}_k; \mathbb{D}_c) + \frac{N}{N_c} \boldsymbol{\epsilon}_c\}_{k=1}^K$ to the server, where each output is a floating-point number and the noise $\boldsymbol{\epsilon}_c$ is negotiated by all clients to be zero-sum. The total uploaded Bytes is $4 \times K$;

**Secure aggregation.** (Lines 15~16, done by server) In order to prevent the server from recovering the exact loss differences and causing privacy leakage (Geiping et al., 2020), we adopt the secure aggregation method (Bonawitz et al., 2017a) that was originally proposed for conventional FL and apply it to BAFFLE. Specifically, all clients negotiate a group of noises $\{\boldsymbol{\epsilon}_c\}_{c=1}^C$ satisfying $\sum_{c=1}^C \boldsymbol{\epsilon}_c = 0$. Then we can reorganize our gradient estimator as

$$\widehat{\nabla_{\mathbf{W}_t}} \mathcal{L}(\mathbf{W}_t) = \frac{1}{K} \sum_{c=1}^C \frac{N_c}{N} \sum_{k=1}^K \left[ \frac{\boldsymbol{\delta}_k}{2\sigma^2} \Delta \mathcal{L}(\mathbf{W}_t, \boldsymbol{\delta}_k; \mathbb{D}_c) \right] = \frac{1}{K} \sum_{k=1}^K \frac{\boldsymbol{\delta}_k}{2\sigma^2} \Delta \mathcal{L}(\mathbf{W}_t, \boldsymbol{\delta}_k),$$

$$\Delta \mathcal{L}(\mathbf{W}_t, \boldsymbol{\delta}_k) = \sum_{c=1}^C \frac{N_c}{N} [\Delta \mathcal{L}(\mathbf{W}_t, \boldsymbol{\delta}_k; \mathbb{D}_c) + \frac{N}{N_c} \boldsymbol{\epsilon}_c]. \tag{7}$$

Since $\{\boldsymbol{\epsilon}_c\}_{c=1}^C$ are zero-sum, there is $\Delta \mathcal{L}(\mathbf{W}_t, \boldsymbol{\delta}_k) = \sum_{c=1}^C \frac{N_c}{N} \Delta \mathcal{L}(\mathbf{W}_t, \boldsymbol{\delta}_k; \mathbb{D}_c)$ and Eq. (7) holds. Therefore, the server can correctly aggregate $\Delta \mathcal{L}(\mathbf{W}_t, \boldsymbol{\delta}_k)$ and protect client privacy against recovering $\Delta \mathcal{L}(\mathbf{W}_t, \boldsymbol{\delta}_k; \mathbb{D}_c)$.

**Remark on communication cost.** After getting the gradient estimation $\widehat{\nabla_{\mathbf{W}_t}} \mathcal{L}(\mathbf{W}_t)$, the server updates the parameters to $\mathbf{W}_{t+1}$ using techniques such as gradient descent with learning rate $\eta$. Similar to the discussion in McMahan et al. (2017), the BAFFLE form presented in Algorithm 1 corresponds to the *batch-level communication (also named FedSGD)* where Lines 11~12 execute once for each round $t$. In batch-level settings, we reduce the uploaded Bytes from $4 \times |\mathbf{W}|$ to $4 \times K$. We can generalize BAFFLE to an analog of *epoch-level communication (also named*

Table 1: The classification accuracy (%) of BAFFLE in **iid scenarios** ($C = 10$) and epoch-level communication settings with different $K$ values ($K_1/K_2$ annotations mean using $K_1$ for MNIST and $K_2$ for CIFAR-10/100). In this configuration, each client updates its local model based on BAFFLE estimated gradients and uploads model updates to the server after an entire epoch on the local dataset. The four guidelines work well under epoch-level settings with total communication rounds 20/40 for MNIST and CIFAR-10/100.

| Settings | | LeNet | | | WRN-10-2 | | |
|---|---|---|---|---|---|---|---|
| | | MNIST | CIFAR-10 | CIFAR-100 | MNIST | CIFAR-10 | CIFAR-100 |
| $K$ | 100/200 | 87.27 | 48.78 | 41.54 | 88.35 | 52.27 | 46.61 |
| | 200/500 | 89.48 | 51.82 | 45.68 | 89.57 | 55.59 | 51.65 |
| | 500/1000 | **92.18** | **53.62** | **48.72** | **95.17** | **58.63** | **53.15** |
| Ablation Study (100/200) | w/o EMA | 85.06 | 47.97 | 36.81 | 85.89 | 50.01 | 45.86 |
| | ReLU | 81.55 | 44.99 | 39.49 | 79.08 | 49.76 | 44.44 |
| | SELU | 86.18 | 48.65 | 37.34 | 76.44 | 43.37 | 41.79 |
| | Central | 76.02 | 45.97 | 36.53 | 77.45 | 42.89 | 39.62 |
| BP Baselines | | 94.31 | 58.75 | 54.67 | 97.11 | 62.29 | 60.08 |

Table 2: The accuracy (%) of BAFFLE in **label non-iid scenarios** ($C = 100$) and epoch-level settings with total comm. rounds 40 and different $K$ values. We employ Dirichlet distribution with $\alpha = 0.3$ to ensure that each client has a unique label distribution.

| Settings | | LeNet | WRN-10-2 |
|---|---|---|---|
| | | CIFAR-10 $\vert$ 100 | CIFAR-10 $\vert$ 100 |
| $K$ | 200 | 35.21 $\vert$ 28.12 | 39.53 $\vert$ 30.44 |
| | 500 | 38.14 $\vert$ 30.92 | 41.69 $\vert$ 32.89 |
| | 1000 | **39.71** $\vert$ **33.35** | **43.42** $\vert$ **34.08** |
| BP Baselines | | 44.41 $\vert$ 38.43 | 51.18 $\vert$ 40.85 |

*FedAvg)*, in which each client updates its local parameters multiple steps using the gradient estimator $\widehat{\nabla_{\mathbf{W}_t}} \mathcal{L}(\mathbf{W}_t, \mathbb{D}_c)$ derived from $\Delta\mathcal{L}(\mathbf{W}_t, \boldsymbol{\delta}_k; \mathbb{D}_c)$ via Eq. (6), and upload model updates to the server after several local epochs. In epoch-level settings, the uploaded Bytes are the same as FedAvg. In experiments, we analyze both batch-level and epoch-level settings for BAFFLE and report the results.

## 3.3 CONVERGENCE ANALYSES

Now we analyze the convergence rate of our gradient estimation method. For continuously differentiable loss functions, we have $\nabla_{\mathbf{W}} \mathcal{L}(\mathbf{W}; \mathbb{D}) = \lim_{\sigma \to 0} \nabla_{\mathbf{W}} \mathcal{L}_\sigma(\mathbf{W}; \mathbb{D})$, so we choose a relatively small value for $\sigma$. The convergence guarantee can be derived as follows:

**Theorem 3.1.** *(Proof in Appendix A) Suppose $\sigma$ is a small value and the central difference scheme in Eq. (2) holds. For perturbations $\{\boldsymbol{\delta}_k\}_{k=1}^K \overset{\text{iid}}{\sim} \mathcal{N}(0, \sigma^2 \mathbf{I})$, the empirical covariance matrix is $\widehat{\boldsymbol{\Sigma}} := \frac{1}{K\sigma^2} \sum_{k=1}^K \boldsymbol{\delta}_k \boldsymbol{\delta}_k^T$ and mean is $\widehat{\boldsymbol{\delta}} := \frac{1}{K} \sum_{k=1}^K \boldsymbol{\delta}_k$. Then for any $\mathbf{W} \in \mathbb{R}^n$, the relation between $\widehat{\nabla_{\mathbf{W}}} \mathcal{L}(\mathbf{W}; \mathbb{D})$ and the true gradient $\nabla_{\mathbf{W}} \mathcal{L}(\mathbf{W}; \mathbb{D})$ can be written as*

$$\widehat{\nabla_{\mathbf{W}}} \mathcal{L}(\mathbf{W}; \mathbb{D}) = \widehat{\boldsymbol{\Sigma}} \nabla_{\mathbf{W}} \mathcal{L}(\mathbf{W}; \mathbb{D}) + o(\widehat{\boldsymbol{\delta}}), \quad (8)$$

*where $\mathbb{E}[\widehat{\boldsymbol{\Sigma}}] = \mathbf{I}$, $\mathbb{E}[\widehat{\boldsymbol{\delta}}] = \mathbf{0}$.*

When expectation is applied to both sides of Eq. (8), we obtain $\mathbb{E}[\widehat{\nabla_{\mathbf{W}}} \mathcal{L}(\mathbf{W}; \mathbb{D})] = \nabla_{\mathbf{W}} \mathcal{L}(\mathbf{W}; \mathbb{D})$, which degrades to Stein's identity. To determine the convergence rate w.r.t. the value of $K$, we have

**Theorem 3.2.** *(Adamczak et al. (2011)) With overwhelming probability, the empirical covariance matrix satisfies the inequality $\|\widehat{\boldsymbol{\Sigma}} - \mathbf{I}\|_2 \leq C_0 \sqrt{\frac{n}{K}}$, where $\|\cdot\|_2$ denotes the 2-norm for matrix and $C_0$ is an absolute positive constant.*

Note that in the finetuning setting, $n$ represents the number of *trainable* parameters, excluding frozen parameters. As concluded, $\widehat{\nabla_{\mathbf{W}}} \mathcal{L}(\mathbf{W}; \mathbb{D})$ provides an unbiased estimation for the true gradients with convergence rate of $\mathcal{O}\left(\sqrt{\frac{n}{K}}\right)$. Empirically, $\widehat{\nabla_{\mathbf{W}}} \mathcal{L}(\mathbf{W}; \mathbb{D})$ is used as a noisy gradient to train models, the generalization of which has been analyzed in previous work (Zhu et al., 2019; Li et al., 2020a).

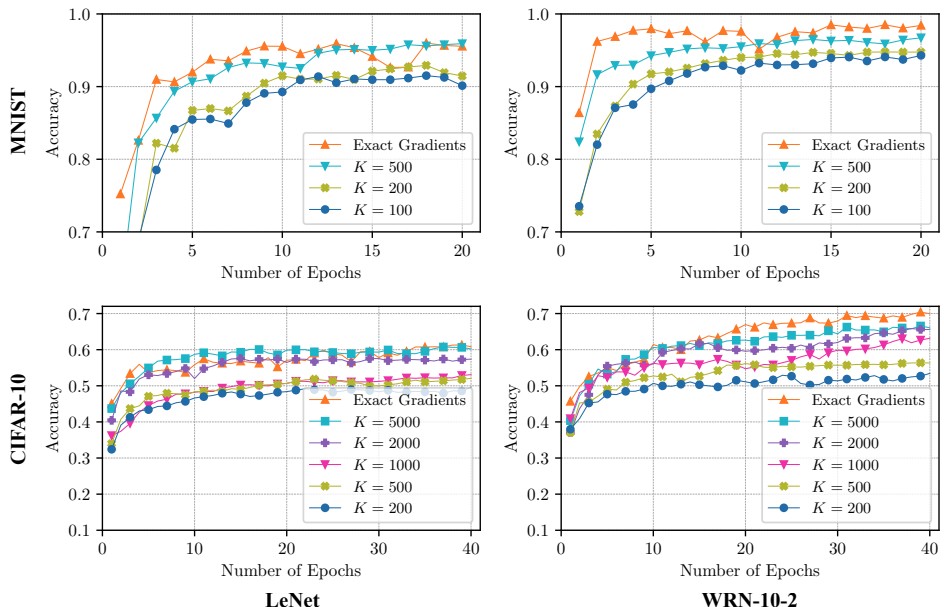

Figure 2: The classification accuracy (%) of BAFFLE in **iid scenarios** ($C = 10$) and batch-level communication settings with various $K$ values. We treat the models trained by exact gradients on conventional FL systems as the backpropagation (BP) baselines. On different datasets and architectures, our BAFFLE achieves comparable performance to the exact gradient results with a reasonable $K$.

## 4 EXPERIMENTS

We evaluate BAFFLE on 4 benchmark datasets: MNIST (LeCun et al., 1998), CIFAR-10/100 (Krizhevsky & Hinton, 2009) and OfficeHome (Venkateswara et al., 2017). We consider three models: 1) LeNet (LeCun et al., 1998) with two convolutional layers as the shallow model ($2.7 \times 10^4$ parameters); 2) WideResNet (Zagoruyko & Komodakis, 2016) with depth = 10 and width = 2 (WRN-10-2) as the light weight deep model ($3.0 \times 10^5$ parameters) and 3) MobileNet (Howard et al., 2017) as the deep neural networks ($1.3 \times 10^7$ parameters) that works on ImageNet.

**Participation and communication settings.** To perform a comprehensive evaluation of BAFFLE, we simulate three popular FL scenarios (Caldas et al., 2018b) with the FedLab (Zeng et al., 2021) participations: *iid participations*, *label non-iid participations* and *feature non-iid participations*. For iid participations, we set the client number $C = 10$ and use uniform distribution to build local datasets. Then we evaluate our BAFFLE on MNIST and CIFAR-10/100 under both *batch-level (FedSGD)* and *epoch-level (FedAvg)* communication settings. For label non-iid participations, we set client number $C = 100$, use Dirichlet distribution with $\alpha = 0.3$ to build clients. For feature non-iid participations, we build clients from the prevailing domain adaptation dataset OfficeHome, which contains 65 categories from 4 different domains, i.e. Art, Clipart, Product and Real-world. We set the total client number to $C = 40$ and generate 10 clients from each domain. As results, we report Top-1 accuracy for MNIST, CIFAR-10 and OfficeHome and Top-5 accuracy for OfficeHome and CIFAR-100.

**Hyperparameters.** Following the settings in Section 2, we use FedAVG to aggregate gradients from multiple clients and use SGD-based optimizer to update global parameters. Specifically, we use Adam (Kingma & Ba, 2015) to train a random initialized model with $\beta = (0.9, 0.99)$, learning rate 0.01 and epochs 20/40 for MNIST and CIFAR-10/100. For OfficeHome, we adapt the transfer learning (Huh et al., 2016) by loading the ImageNet-pretrained model and finetuning the final layers with Adam, but setting learning rate 0.005 and epochs 40. In BAFFLE, the perturbation scale $\sigma$ and number $K$ are the most important hyperparameters. As shown in Theorem 3.1, with less noise and more samples, the BAFFLE will obtain more accurate gradients, leading to improved performance. However, there exists a trade-off between accuracy and computational efficiency: an extremely small $\sigma$ will cause the underflow problem (Goodfellow et al., 2016) and a large $K$ will increase computational cost. In practice, we empirically set $\sigma = 10^{-4}$ because it is the smallest value that does not cause numerical problems in all experiments, and works well on edge devices with half-precision floating-point numbers. We also evaluate the impact of $K$ across a broad range from 100 to 5000.

Table 3: The Top-1|Top-5 accuracy (%) of BAFFLE on OfficeHome with **feature non-iid partici-pations** ($C = 40$) and epoch-level settings with total comm. rounds 40. We utilize the pretrained MobileNet, freeze the backbone and finetune the classification layers.

| Settings | | Domains | | | | Avg. |
|---|---|---|---|---|---|---|
| | | Art | Clipart | Product | Real World | |
| $K$ | 20 | 44.75\|69.46 | 52.48\|73.88 | 66.63\|89.77 | 63.78\|87.83 | 56.91\|80.24 |
| | 50 | 47.87\|71.32 | 53.43\|76.83 | 71.28\|91.74 | 67.02\|89.95 | 59.90\|82.46 |
| | 100 | 50.32\|74.42 | 57.43\|80.73 | 74.19\|93.02 | 69.53\|90.43 | 62.87\|84.65 |
| | 200 | 51.42\|76.64 | 60.98\|86.41 | 76.05\|94.42 | 71.51\|93.14 | 64.98\|87.65 |
| | 500 | **53.33\|77.85** | **62.58\|86.64** | **78.84\|95.23** | **73.17\|93.85** | **66.98\|88.40** |
| BP Baselines | | 55.71\|80.43 | 65.13\|88.65 | 82.44\|96.05 | 77.19\|95.04 | 70.12\|90.04 |

## 4.1 FOUR GUIDELINES FOR BAFFLE

For a general family of continuously differentiable models, we analyze their convergence rate of BAFFLE in Section 3.3. Since deep networks are usually stacked with multiple linear layers and non-linear activation, this layer linearity can be utilized to improve the accuracy-efficiency trade-off. Combining the linearity property and the unique conditions in edge devices (e.g., small data size and half-precision format), we present four guidelines for model design and training that can increase accuracy without introducing extra computation (Appendix D shows the details of linearity analysis):

**Using twice forward difference (twice-FD) scheme rather than central scheme.** Combining difference scheme Eq. (1) and Eq. (2), we find that by executing twice as many forward inferences (i.e. $\mathbf{W} \pm \boldsymbol{\delta}$), the central scheme achieves lower residuals than twice-FD, despite the fact that twice-FD can benefit from additional sample times. With the same forward times (e.g., $2K$), determining which scheme performs better is a practical issue. As shown in Appendix D, we find that twice-FD performs better in all experiments, in part because the linearity reduces the benefit from second-order residuals.

**Using Hardswish in BAFFLE.** ReLU is effective when the middle features ($h(\cdot)$ denotes the feature mapping) have the same sign before and after perturbations, i.e. $h(\mathbf{W} + \boldsymbol{\delta}) \cdot h(\mathbf{W}) > 0$. Since ReLU is not differentiable at zero, the value jump occurs when the sign of features changes after perturbations, i.e. $h(\mathbf{W} + \boldsymbol{\delta}) \cdot h(\mathbf{W}) < 0$. We use Hardswish (Howard et al., 2019) to overcome this problem as it is continuously differentiable at zero and easy to implement on edge devices.

**Using exponential moving average (EMA) to reduce oscillations.** As shown in Theorem 3.1, there exists an zero-mean white-noise $\widehat{\boldsymbol{\delta}}$ between the true gradient and our estimation. To smooth out the oscillations caused by white noise, we apply EMA strategies from BYOL (Grill et al., 2020) to the global parameters, with a smoothing coefficient of 0.995.

**Using GroupNorm as opposed to BatchNorm.** On edge devices, the dataset size is typically small, which leads to inaccurate batch statistics estimation and degrades performance when using BatchNorm. Thus we employ GroupNorm (Wu & He, 2020) to solve this issue.

## 4.2 PERFORMANCE ON IID CLIENTS

Following the settings in Section 4.1, we evaluate the performance of BAFFLE in the iid scenarios. We reproduce all experiments on the BP-based FL systems with the same settings and use them as the baselines. We refer to the baseline results as *exact gradients* and report the training process of BAFFLE in Figure 2. The value of $K$ (e.g., 200 for LeNet and 500 for WRN-10-2) is much less than the dimensions of parameter space (e.g., $2.7 \times 10^4$ for LeNet and $3 \times 10^5$ for WRN-10-2). Since the convergence rate to the exact gradient is $\mathcal{O}\left(\sqrt{\frac{n}{K}}\right)$, the marginal benefit of increasing $K$ decreases. For instance, increasing $K$ from 2000 to 5000 on CIFAR-10 with WRN-10-2 barely improves accuracy by 2%. Given that the convergence rate of Gaussian perturbations is $\mathcal{O}\left(\sqrt{\frac{n}{K}}\right)$, the sampling efficiency may be improved by choosing an alternative distribution for perturbations.

**Ablation studies.** As depicted in Figure 3, we conduct ablation studies for BAFFLE to evaluate the aforementioned guidelines. In general, twice-FD, Hardswish and EMA can all improve the accuracy. For two difference schemes, we compare the twice-FD to central scheme with the same computation cost and show that the former outperforms the later, demonstrating that linearity reduces the gain

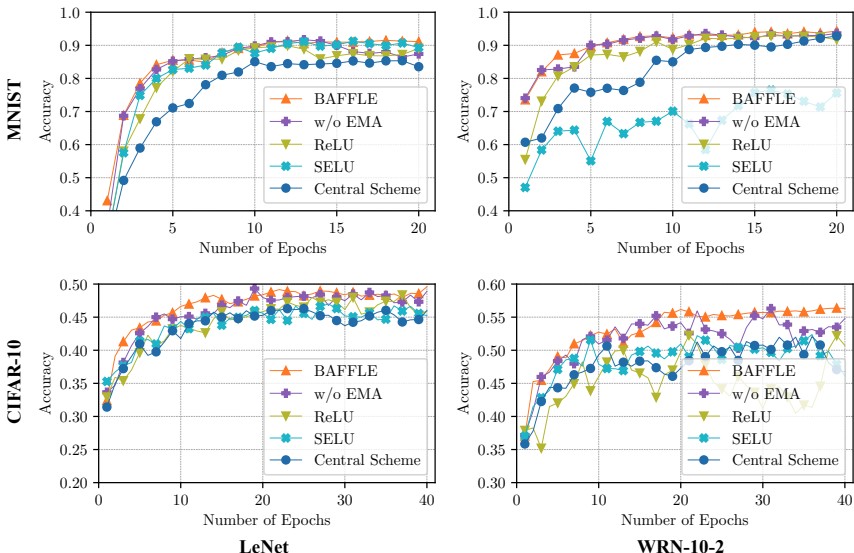

Figure 3: The ablation study of BAFFLE guidelines, with $K = 100$ on MNIST and $K = 500$ on CIFAR-10. As seen, twice-FD, Hardswish, and EMA all improve performance without extra computation. EMA reduces oscillations by lessening Gaussian noise.

from second-order residuals. As to activation functions, Hardswish is superior to ReLU and SELU because it is differentiable at zero and vanishes to zero in the negative part. Moreover, EMA enhances the performance of training strategies by reducing the effect of white noise.

**Communication efficiency.** Compared to the *batch-level communication settings (FedSGD)* in a BP-based FL system, BAFFLE requires each client to upload a $K$-dimensional vector to the server and downloads the updated global parameters in each communication round. Since $K$ is significantly less than the parameter amounts (e.g., 500 versus 0.3 million), BAFFLE reduces data transfer by approximately half. To reduce communication costs, the prevalent FL system requires each client to perform model optimization on the local training dataset and upload the model updates to the server after a specified number of local epochs. BAFFLE can also perform *epoch-level communications* by employing an $\mathcal{O}(n)$ additional memory to store the perturbation in each forward and estimate the local gradient using Eq. (6). Then each client optimizes the local model with SGD and uploads local updates after several epochs. As shown in Table 1, we evaluate the performance of BAFFLE under one-epoch communication settings. As epoch-level communication is more prevalent in the real-world FL, all the following experiments will be conducted in this context. In brief, BAFFLE uploads the same Bytes as BP-based FL in epoch-level comm. while the total comm. rounds are much less than FedZO (Fang et al., 2022), e.g. 20 versus 1000 on MNIST.

### 4.3 PERFORMANCE ON NON-IID CLIENTS

Following Section 4.1, we evaluate the performance of BAFFLE in both label non-iid and feature non-iid scenarios. **For label non-iid scenarios**, we use the CIFAR-10/100 datasets and employ Dirichlet distribution to ensure that each client has a unique label distribution. We evaluate the performance of BAFFLE with 100 clients and various K values. As seen in Table 2, the model suffers a significant drop in accuracy (e.g., $14\%$ in CIFAR-10 and $16\%$ in CIFAR-100) due to the label non-iid effect. **For feature non-iid scenarios**, we construct clients using the OfficeHome dataset and use MobileNet as the deep model. As seen in Table 3, we use the transfer learning strategy to train MobileNet, i.e., we load the parameters pretrained on ImageNet, freeze the backbone parameters, and retrain the classification layers. The accuracy decrease is approximately $3\% \sim 5\%$.

### 4.4 COMPUTATION EFFICIENCY, MEMORY AND ROBUSTNESS

BAFFLE uses $K$ times forward passes instead of backward. Since the backward pass is about as expensive as two normal forward passes (Hinton & Srivastava, 2010) and five single-precision accelerated forward passes Nakandala et al. (2020), BAFFLE results in approximately $\frac{K}{5}$ times the

Table 4: The GPU memory cost (MB) of vanilla backpropagation and BAFFLE, respectively. 'min~max' denotes the minimum and maximum dynamic memory requirements for BAFFLE. We also report the ratio (%) of vanilla BP to BAFFLE's max memory cost.

| Backbone | CIFAR-10/100 | | | OfficeHome/ImageNet | | |
|---|---|---|---|---|---|---|
| | BP | BAFFLE | Ratio | BP | BAFFLE | Ratio |
| LeNet | 1680 | 67~174 | **10.35** | 2527 | 86~201 | **7.95** |
| WRN-10-2 | 1878 | 75~196 | **10.43** | 3425 | 94~251 | **7.32** |
| MobileNet | 2041 | 102~217 | **10.63** | 5271 | 121~289 | **5.48** |

computation expense of BP-based FL. Although BAFFLE results in $\frac{K}{5} - 1$ times extra computation cost, we show the cost can be reduced with proper training strategies, e.g., the transfer learning in Table 3 can reduce $K$ to 20 on the MobileNet and the $224 \times 224$ sized OfficeHome dataset.

Moreover, BAFFLE can reduce huge memory cost on edge devices with the efficiency in static memory and dynamic memory. The auto-differential framework is used to run BP on deep networks, which requires extra static memory (e.g., 200MB for Caffe (Jia et al., 2014) and 1GB for Pytorch (Paszke et al., 2019a)) and imposes a considerable burden on edge devices such as IoT sensors. Due to the necessity of restoring intermediate states, BP also requires enormous amounts of dynamic memory ($\geq$ 5GB for MobileNet (Gao et al., 2020)). Since BAFFLE only requires inference, we can slice the computation graph and execute the forwards per layer (Kim et al., 2020). As shown in Table 4, BAFFLE reduces the memory cost to 5%~10% by executing layer-by-layer inference. By applying kernel-wise computations, we can further reduce the memory cost to approximately 1% (e.g., 64MB for MobileNet (Truong et al., 2021)), which is suitable for scenarios with extremely limited storage resources, such as TEE.

Recent works exploit TEE to protect models from white-box attacks by preventing model exposure (Kim et al., 2020). However, due to the security guarantee, the usable memory of TEE is usually small (Truong et al., 2021) (e.g., 90MB on Intel SGX for Skylake CPU (McKeen et al., 2016)), which is typically far less than what a backpropagation-based FL system requires. In contrast, BAFFLE can execute in TEE due to its little memory cost (more details are in Appendix C). Membership inference attacks and model inversion attacks need to repeatedly perform model inference and obtain confidence values or classification scores (Shokri et al., 2017; Zhang et al., 2020). Given that BAFFLE provides stochastic loss differences $\Delta\mathcal{L}(\mathbf{W}, \boldsymbol{\delta}; \mathbb{D})$ associated with the random perturbation $\boldsymbol{\delta}$, the off-the-shelf inference attacks may not perform on BAFFLE directly (while adaptively designed attacking strategies are possible to evade BAFFLE). Motivated by differential privacy (Abadi et al., 2016), we further design heuristic experiments to study the information leakage from $\Delta\mathcal{L}$ (details in Appendix E). As shown in Figure 5, the $\Delta\mathcal{L}$ between real data and random noise is hard to distinguish, indicating it is difficult for attackers to obtain useful information from BAFFLE's outputs.

## 5 CONCLUSION AND DISCUSSION

Backpropagation is the gold standard for training deep networks, and it is also utilized by traditional FL systems. However, backpropagation is unsuited for edge devices due to their limited resources and possible lack of reliability. Using zero-order optimization techniques, we explore the possibility of BAFFLE in this paper. We need to specify that there are scenarios in which clients are fully trusted and have sufficient computing and storage resources. In these situations, traditional FL with backpropagation is preferred.

While our preliminary studies on BAFFLE have generated encouraging results, there are still a number of tough topics to investigate: **(i)** Compared to the models trained using exact gradients, the accuracy of models trained using BAFFLE is inferior. One reason is that we select small values of $K$ (e.g., 500) relative to the number of model parameters (e.g., $3.0 \times 10^5$); another reason is that gradient descent is designed for exact gradients, whereas our noisy gradient estimation may require advanced learning algorithms. **(ii)** The empirical variance of zero-order gradient estimators affects training convergence in BAFFLE. It is crucial to research variance reduction approaches, such as control variates and non-Gaussian sampling distributions. **(iii)** Stein's identity is proposed for loss functions with Gaussian noises imposed on model parameters. Intuitively, this smoothness is related to differential privacy in FL, but determining their relationship requires theoretical derivations.

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

## A PROOFS

### A.1 PROOF OF STEIN'S IDENTITY

We recap the proof of Stein's identity following He et al. (2022), where

$$
\begin{aligned}
\nabla_{\mathbf{W}} \mathcal{L}_\sigma(\mathbf{W}; \mathbb{D}) &= \nabla_{\mathbf{W}} \mathbb{E}_{\boldsymbol{\delta} \sim \mathcal{N}(0, \sigma^2 I)} \mathcal{L}(\mathbf{W} + \boldsymbol{\delta}; \mathbb{D}) \\
&= (2\pi)^{-\frac{n}{2}} \cdot \nabla_{\mathbf{W}} \int \mathcal{L}(\mathbf{W} + \boldsymbol{\delta}; \mathbb{D}) \cdot \exp\left(-\frac{\|\boldsymbol{\delta}\|_2^2}{2\sigma^2}\right) d\boldsymbol{\delta} \\
&= (2\pi)^{-\frac{n}{2}} \cdot \int \mathcal{L}(\widetilde{\mathbf{W}}; \mathbb{D}) \cdot \nabla_{\mathbf{W}} \exp\left(-\frac{\|\widetilde{\mathbf{W}} - \mathbf{W}\|_2^2}{2\sigma^2}\right) d\widetilde{\mathbf{W}} \\
&= (2\pi)^{-\frac{n}{2}} \cdot \int \mathcal{L}(\mathbf{W} + \boldsymbol{\delta}; \mathbb{D}) \cdot \frac{\boldsymbol{\delta}}{\sigma^2} \cdot \exp\left(-\frac{\|\boldsymbol{\delta}\|_2^2}{2\sigma^2}\right) d\boldsymbol{\delta} \\
&= \mathbb{E}_{\boldsymbol{\delta} \sim \mathcal{N}(0, \sigma^2 \mathbf{I})} \left[\frac{\boldsymbol{\delta}}{\sigma^2} \mathcal{L}(\mathbf{W} + \boldsymbol{\delta}; \mathbb{D})\right].
\end{aligned}
\tag{9}
$$

By symmetry, we change $\boldsymbol{\delta}$ to $-\boldsymbol{\delta}$ and obtain

$$
\nabla_{\mathbf{W}} \mathcal{L}_\sigma(\mathbf{W}; \mathbb{D}) = -\mathbb{E}_{\boldsymbol{\delta} \sim \mathcal{N}(0, \sigma^2 \mathbf{I})} \left[\frac{\boldsymbol{\delta}}{\sigma^2} \mathcal{L}(\mathbf{W} - \boldsymbol{\delta}; \mathbb{D})\right],
\tag{10}
$$

and further we prove that

$$
\begin{aligned}
\nabla_{\mathbf{W}} \mathcal{L}_\sigma(\mathbf{W}; \mathbb{D}) &= \frac{1}{2} \mathbb{E}_{\boldsymbol{\delta} \sim \mathcal{N}(0, \sigma^2 \mathbf{I})} \left[\frac{\boldsymbol{\delta}}{\sigma^2} \mathcal{L}(\mathbf{W} + \boldsymbol{\delta}; \mathbb{D})\right] - \frac{1}{2} \mathbb{E}_{\boldsymbol{\delta} \sim \mathcal{N}(0, \sigma^2 \mathbf{I})} \left[\frac{\boldsymbol{\delta}}{\sigma^2} \mathcal{L}(\mathbf{W} - \boldsymbol{\delta}; \mathbb{D})\right] \\
&= \mathbb{E}_{\boldsymbol{\delta} \sim \mathcal{N}(0, \sigma^2 \mathbf{I})} \left[\frac{\boldsymbol{\delta}}{2\sigma^2} \Delta\mathcal{L}(\mathbf{W}, \boldsymbol{\delta}; \mathbb{D})\right].
\end{aligned}
$$

$\square$

### A.2 PROOF OF THEOREM 3.1

We rewrite the format of $\widehat{\nabla_{\mathbf{W}}} \mathcal{L}(\mathbf{W}; \mathbb{D})$ as follows:

$$
\begin{aligned}
\widehat{\nabla_{\mathbf{W}}} \mathcal{L}(\mathbf{W}; \mathbb{D}) &= \frac{1}{K} \sum_{k=1}^{K} [\frac{\boldsymbol{\delta}_k}{2\sigma^2} \Delta\mathcal{L}(\mathbf{W}, \boldsymbol{\delta}_k; \mathbb{D})] \\
&= \frac{1}{K} \sum_{k=1}^{K} [\frac{\boldsymbol{\delta}_k}{2\sigma^2} \left(2\boldsymbol{\delta}_k^\top \nabla_{\mathbf{W}} \mathcal{L}(\mathbf{W}; \mathbb{D}) + o(\|\boldsymbol{\delta}_k\|_2^2)\right)] \quad \text{(using central scheme in Eq. (2))} \\
&= \frac{1}{K\sigma^2} \sum_{k=1}^{K} [\boldsymbol{\delta}_k \boldsymbol{\delta}_k^\top] \nabla_{\mathbf{W}} \mathcal{L}(\mathbf{W}; \mathbb{D}) + \frac{1}{K} \sum_{k=1}^{K} \frac{\boldsymbol{\delta}_k}{2\sigma^2} o(\|\boldsymbol{\delta}_k\|_2^2) \\
&= \widehat{\boldsymbol{\Sigma}} \nabla_{\mathbf{W}} \mathcal{L}(\mathbf{W}; \mathbb{D}) + \frac{1}{K} \sum_{k=1}^{K} \frac{\boldsymbol{\delta}_k}{2\sigma^2} o(\|\boldsymbol{\delta}_k\|_2^2).
\end{aligned}
\tag{11}
$$

Then we prove $\frac{1}{K} \sum_{k=1}^{K} \frac{\boldsymbol{\delta}_k}{2\sigma^2} o(\|\boldsymbol{\delta}_k\|_2^2) = o(\widehat{\boldsymbol{\delta}})$. Suppose $\boldsymbol{\delta}_k = (\delta_{k,1}, \cdots, \delta_{k,n})$, then we have $\frac{\|\boldsymbol{\delta}_k\|_2^2}{\sigma^2} = \sum_{i=1}^{n} (\frac{\delta_{k,i}}{\sigma})^2$. Since $\forall i, \frac{\delta_{k,i}}{\sigma} \sim \mathcal{N}(0,1)$, we have $\frac{\|\boldsymbol{\delta}_k\|_2^2}{\sigma^2} \sim \chi^2(n)$ and $\mathbb{E}(\frac{\|\boldsymbol{\delta}_k\|_2^2}{\sigma^2}) = n$. So with high probability, $\frac{o(\|\boldsymbol{\delta}_k\|_2^2)}{\sigma^2} = o(n)$. Substituting it into Eq. (11), we have with high probability,

$$
\frac{1}{K} \sum_{k=1}^{K} \frac{\boldsymbol{\delta}_k}{2\sigma^2} o(\|\boldsymbol{\delta}_k\|_2^2) = \widehat{\boldsymbol{\delta}} \cdot o(n) = o(\widehat{\boldsymbol{\delta}}),
$$

where we regard $n$ as a constant for a given model architecture. Finally, we prove $\mathbb{E}[\widehat{\boldsymbol{\delta}}] = \mathbf{0}$ and $\mathbb{E}[\widehat{\boldsymbol{\Sigma}}] = \mathbf{I}$. It is trivial that $\mathbb{E}[\widehat{\boldsymbol{\delta}}] = \mathbf{0}$ since $\widehat{\boldsymbol{\delta}} \sim \mathcal{N}(0, \frac{1}{K\sigma^2} \mathbf{I})$. For $\mathbb{E}[\widehat{\boldsymbol{\Sigma}}] = \mathbf{I}$, we can observe by

Figure 4: A sketch map to run BAFFLE in one trusted execution environment. The pipeline contains three steps: (1) Load the data and model into the security storage. (2) Load the code of BAFFLE into the root of trust. (3) Run the BAFFLE program in a separation kernel.

examining each of its entries

$$\widehat{\boldsymbol{\Sigma}}_{[ij]} = \frac{1}{K\sigma^2} \sum_{k=1}^{K} \delta_{k[i]}\delta_{k[j]} = \frac{1}{K} \sum_{k=1}^{K} \frac{\delta_{k[i]}}{\sigma} \frac{\delta_{k[j]}}{\sigma}, \tag{12}$$

where we have used subscripts $[ij]$ and $[i]$ to denote the usual indexing of matrices and vectors. Specifically, for diagonal entries (i.e., $i = j$), we observe $K \cdot \widehat{\boldsymbol{\Sigma}}_{[ii]} = \sum_{k=1}^{K} \left(\frac{\delta_{k[i]}}{\sigma}\right)^2$ distributes as $\chi^2(K)$, which means $\mathbb{E}[\widehat{\boldsymbol{\Sigma}}_{[ii]}] = 1 = \mathbf{I}_{[ii]}$ and $\text{Var}[\widehat{\boldsymbol{\Sigma}}_{[ii]}] = \frac{2}{K}$; for non-diagonal entries (i.e., $i \neq j$), we have $\mathbb{E}[\widehat{\boldsymbol{\Sigma}}_{[ij]}] = \frac{1}{K}\sum_{k=1}^{K} \mathbb{E}\left[\frac{\delta_{k[i]}}{\sigma}\frac{\delta_{k[j]}}{\sigma}\right] = \frac{1}{K}\sum_{k=1}^{K} \frac{\mathbb{E}[\delta_{k[i]}]}{\sigma}\frac{\mathbb{E}[\delta_{k[j]}]}{\sigma} = 0 = \mathbf{I}_{[ij]}$, due to the independence between different dimensions in $\boldsymbol{\delta}_k$. $\qquad\square$

## B  RELATED WORK

Along the research routine of FL, many efforts have been devoted to, e.g., dealing with non-IID distributions (Zhao et al., 2018; Sattler et al., 2019; Eichner et al., 2019; Wang et al., 2020b; Li et al., 2020c), multi-task learning (Smith et al., 2017; Marfoq et al., 2021), and preserving privacy of clients (Bonawitz et al., 2016; 2017b; McMahan et al., 2018; Truex et al., 2019; Hao et al., 2019; Lyu et al., 2022; Ghazi et al., 2020; Liu et al., 2020b). Below we introduce the work on efficiency and vulnerability in FL following the survey of Kairouz et al. (2021), which is more related to this paper.

**Efficiency in FL.** It is widely understood that the communication and computational efficiency is a primary bottleneck for deploying FL in practice (Wang et al., 2019b; Rothchild et al., 2020; Chen et al., 2021; Balakrishnan et al., 2022; Wang et al., 2022). Specifically, communicating between the server and clients could be potentially expensive and unreliable. The seminal work of Konečnỳ et al. (2016) introduces sparsification and quantization to reduce the communication cost, where several theoretical works investigate the optimal trade-off between the communication cost and model accuracy (Zhang et al., 2013; Braverman et al., 2016; Han et al., 2018; Acharya et al., 2020; Barnes et al., 2020). Since practical clients usually have slower upload than download bandwidth, much research interest focuses on gradient compression (Suresh et al., 2017; Alistarh et al., 2017; Horváth et al., 2019; Basu et al., 2019). On the other hand, different methods have been proposed to reduce the computational burden of local clients (Caldas et al., 2018a; Hamer et al., 2020; He et al., 2020), since these clients are usually edge devices with limited resources. Training paradigms exploiting tensor factorization in FL can also achieve promising performance (Kim et al., 2017; Ma et al., 2019).

**Vulnerability in FL.** The characteristic of decentralization in FL is beneficial to protecting data privacy of clients, but in the meanwhile, providing white-box accessibility of model status leaves flexibility for malicious clients to perform poisoning/backdoor attacks (Bhagoji et al., 2019; Bagdasaryan et al., 2020; Wang et al., 2020a; Xie et al., 2020; Pang et al., 2021), model/gradient inversion attacks (Zhang et al., 2020; Geiping et al., 2020; Huang et al., 2021), and membership inference attacks (Shokri et al., 2017; Nasr et al., 2019; Luo et al., 2021). To alleviate the vulnerability in FL, several defense strategies have been proposed via selecting reliable clients (Kang et al., 2020), data augmentation (Borgnia et al., 2021), update clipping (Sun et al., 2019), robust training (Li et al.,

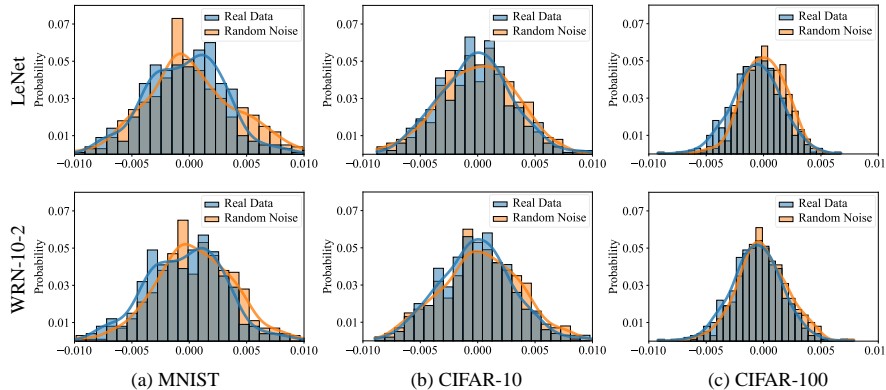

(a) MNIST         (b) CIFAR-10         (c) CIFAR-100

Figure 5: The robustness of BAFFLE to inference attacks. For real data, we randomly sample some input-label pairs from the validation dataset. For random noise, we generate input-label pairs from standard normal distribution. We sample 500 perturbations $\boldsymbol{\delta}$ from $\mathcal{N}(0, \sigma^2 \mathbf{I})$, collect the values of $\Delta \mathcal{L}(\mathbf{W}, \boldsymbol{\delta}; \mathbb{D})$ for real data and random noise separately, and compare their distributions.

2021), model perturbation (Yang et al., 2022), detection methods (Seetharaman et al., 2021; Dong et al., 2021), and methods based on differential privacy (Wei et al., 2020).

## C   TRUSTED EXECUTION ENVIRONMENT

A trusted execution environment (TEE) (Sabt et al., 2015) is regarded as the ultimate solution for defending against all white-box attacks by preventing any model exposure. TEE protects both data and model security with three components: physical secure storage to ensure the confidentiality, integrity, and tamper-resistance of stored data; a root of trust to load trusted code; and a separate kernel to execute code in an isolated environment, as illustrated in Figure 4. Using TEE, the FL system is able to train deep models without revealing model specifics. However, due to the security guarantee, the usable memory of TEE is typically small (Truong et al., 2021) (e.g., 90MB on Intel SGX for Skylake CPU (McKeen et al., 2016)), which is considerably less than what deep models require for backpropagation (e.g., $\geq$ 5GB for VGG-16 (Gao et al., 2020)).

## D   CONVERGENCE ANALYSES OF DEEP LINEAR NETWORKS IN BAFFLE

We analyze the convergence of BAFFLE in Section 3 using a general technique applicable to any continuously differentiable models corresponding to the loss function $\mathcal{L}(\mathbf{W}; \mathbb{D})$. Since deep networks are the most prevalent models in FL, which has strong linearity, it is simpler to investigate the convergence of deep linear networks (Saxe et al., 2013).

Consider a two-layer deep linear network in a classification task with $L$ categories. We denote the model parameters as $\{\mathbf{W}_1, \mathbf{W}_2\}$, where in the first layer $\mathbf{W}_1 \in \mathbb{R}^{n \times m}$, in the second layer $\mathbf{W}_2 \in \mathbb{R}^{L \times n}$ consists of $L$ vectors related to the $L$ categories as $\{\mathbf{w}_2^l\}_{l=1}^L$ and $\mathbf{w}_2^c \in \mathbb{R}^{1 \times n}$. For the input data $\mathbf{X} \in \mathbb{R}^{m \times 1}$ with label $y$, we train the deep linear network by maximizing the classification score on the $y$-th class. Since there is no non-linear activation in deep linear networks, the forward inference can be represented as $h = \mathbf{w}_2^y \mathbf{W}_1 \mathbf{X}$, and the loss is $-h$. It is easy to show that $\frac{\partial h}{\partial \mathbf{w}_2^y} = (\mathbf{W}_1 \mathbf{X})^\top$ and $\frac{\partial h}{\partial \mathbf{W}_1} = (\mathbf{X} \mathbf{w}_2^y)^\top$. We sample $\boldsymbol{\delta}_1, \boldsymbol{\delta}_2$ from noise generator $\mathcal{N}(0, \sigma^2 \mathbf{I})$, where $\boldsymbol{\delta}_1 \in \mathbb{R}^{n \times m}$ and $\boldsymbol{\delta}_2 \in \mathbb{R}^{1 \times n}$. Let $h(\boldsymbol{\delta_1}, \boldsymbol{\delta_2}) := (\mathbf{w}_2^y + \boldsymbol{\delta}_2)(\mathbf{W}_1 + \boldsymbol{\delta}_1)\mathbf{X}$, we discover that the BAFFLE estimation in Eq. (6) follows the same pattern for both forward (2) and central schemes (3):

$$
\begin{aligned}
\Delta_{\text{for}} h(\boldsymbol{\delta}_1, \boldsymbol{\delta}_2) &:= h(\boldsymbol{\delta}_1, \boldsymbol{\delta}_2) - h(\mathbf{0}, \mathbf{0}); \\
\Delta_{\text{ctr}} h(\boldsymbol{\delta}_1, \boldsymbol{\delta}_2) &:= h(\boldsymbol{\delta}_1, \boldsymbol{\delta}_2) - h(-\boldsymbol{\delta}_1, -\boldsymbol{\delta}_2); \\
\frac{\Delta_{\text{for}} h(\boldsymbol{\delta}_1, \boldsymbol{\delta}_2)}{\sigma^2} &= \frac{\Delta_{\text{ctr}} h(\boldsymbol{\delta}_1, \boldsymbol{\delta}_2)}{2\sigma^2} = \frac{1}{\sigma^2}(\mathbf{w}_2^c \boldsymbol{\delta}_1 \mathbf{X} + \boldsymbol{\delta}_2 \mathbf{W}_1 \mathbf{X}).
\end{aligned}
\tag{13}
$$

This equivalent form in deep linear networks illustrates that the residual benefit from the central scheme is reduced by the linearity, hence the performance of the two finite difference schemes

described above is same in deep linear networks. We refer to this characteristic as FD scheme independence. We also find the property of $\sigma$ independence, that is, the choice of $\sigma$ does not effect the results of finite difference, due to the fact that $\frac{\delta_1}{\sigma}$ and $\frac{\delta_2}{\sigma}$ follow the standard normal distribution.

Based on the findings from Eq. (13), we propose the following useful guideline that improves accuracy under the same computation cost: *Using twice forward difference (twice-FD) scheme rather than central scheme.* Combining the forward scheme Eq. (2) and central scheme Eq. (3), we find that the central scheme produces smaller residuals than the forward scheme by executing twice as many forward inferences, i.e. $\mathbf{W} \pm \boldsymbol{\delta}$. With the same forward inference times (e.g., $2K$), one practical difficulty is to identify which scheme performs better. We find that the forward scheme performs better in all experiments, in part because the linearity reduces the benefit from second-order residuals, as demonstrated by Eq. (13).

## E  ROBUSTNESS TO INFERENCE ATTACKS

To explore the information leakage from outputs $\Delta\mathcal{L}$, we design heuristic experiments. Regular attacks such as membership inference attacks and model inversion attacks cannot directly target BAFFLE since they must repeatedly do model inference and get confidence values or classification scores. To analyze the possibility of information leaking, we employ the concept of differential privacy (Abadi et al., 2016) and compare the BAFFLE's outputs from private data to random noise. If we cannot discriminate between private data and random noise merely from the BAFFLE's outputs, we can assert that the outputs do not contain private information. In details, we utilize the validation dataset as the private data and generate random input pairs from Gaussian and Laplacian noise as $(\tilde{\mathbf{X}}, \tilde{y})$. Then we apply BAFFLE to both private data and random noise and compare the distributions of their respective outputs $\Delta\mathcal{L}$. As shown in Figure 5, it is difficult to distinguish the BAFFLE's outputs between private data and random noise, showing that it is difficult for attackers to acquire meaningful information rather than random noise from the BAFFLE's outputs.

