# OpenReview forum: "BAFFLE: A Baseline of Backpropagation-Free Federated Learning"
_ICLR.cc/2024/Conference — Submitted to ICLR 2024_

### Official Review · Reviewer_9br4 · 2023-10-27

**Soundness:** 2 fair
**Presentation:** 3 good
**Contribution:** 2 fair
**Rating:** 3
**Confidence:** 4

**Summary:**

This work proposes BAFFLE, a method for zero-order optimization in the federated setting. The main motivation behind zero-order optimization is its memory efficiency, as no activations need to be stored for a backward pass, it is compatible with hardware designed for inference only and, furthermore, its memory efficiency allows for training within a trusted execution environment (TEE). The recipe is quite simple, in that the server communicates a random seed and a model to the clients and then each of the clients does $2K$ forward passes with weights $w \pm \delta_k$ and their local data.  The loss differences are then communicated and aggregated at the server where, together with the perturbations $\delta_k$, an approximation to the model gradient can be computed. The authors provide guidelines that allow such a scheme to be successful and then show, for moderately large values of $K$, that BAFFLE can reach performance comparable to vanilla backpropagation.

**Strengths:**

The paper overall attempts to tackle an interesting problem that is useful in practice, especially given the success of large models. It is well written and clear in the description of the method and motivation. As for other strengths:
- The memory efficiency of BAFFLE is especially useful in the context of training large models on resource constrained devices. This memory efficiency also allows for training within a TEE.
- The zero-order nature might allow BAFFLE to be compatible with inference only models and accelerators.
- The authors also propose useful guidelines for BAFFLE, which I believe can be useful for zero-order optimization in general.
- The authors provide a theoretical convergence analysis of BAFFLE.
- Experiments cover a decent variety of settings, from different settings of data non-i.i.d.-ness to computation / memory efficiency and robustness

**Weaknesses:**

I believe this work has relatively limited novelty, especially given prior work. More specifically, the authors discuss both forward finite differences and central finite differences but in the experiments they mention using forward finite differences, due to it being better for the specific amount of sample budget they consider. As far as I see, forward finite differences is essentially the exact same thing as Evolution Strategies (e.g., see [1]) which has already been considered for training neural network type of models in parallel (again, see [1]).

Furthermore, some of the claims of the method could use more details. For example, the authors mention that BAFFLE can apply to models optimised for inference, e.g., models that have been quantised and pruned. However, adding Gaussian noise to the weights can break both the quantization and (unstructured) pruning, so it is unclear whether these models would be able to run on their dedicated accelerators (which sometimes require weights to, e.g., be properly qunatized in a uniform grid). The authors also argue that secure aggregation (SAG) is compatible with BAFFLE, however, as far as I know, SAG requires the inputs to be quantised and this is not discussed in the main text.

Finally, if memory constraint is an issue, I believe an important baseline is missing and that is gradient checkpointing for BP, which can also reduce memory costs at the expense of more compute (which is the tradeoff BAFFLE also has).

[1] Evolution Strategies as a Scalable Alternative to Reinforcement Learning, Salimans et al., 2017

**Questions:**

I have the following questions and suggestions
- Do the authors quantise the loss differences in order to allow for SAG? If yes, what is the scheme? How does it affect the convergence behaviour?
- The convergence based on a finite number of forward passes depends on a constant $C_0$. How large is that constant in practice?
- The harms of batch normalisation and its replacement by group normalisation, that the authors argue for as a guideline, is not new and has been discussed in various other works in federated learning, e.g, [2]. I would suggest that the authors properly attribute these prior works.
- Are the experiments shown with full client participation? Partial client participation is more common in cross-device settings (which is usually the case where resource constraints appear), so it would be interesting to see results with that, especially given that it increases gradient variance.
- I would suggest to the authors to elaborate on any differences with respect to [1].
- I would suggest to the authors to elaborate about the feasibility of evaluating quantized / pruned models with Gaussian noise on their weights on dedicated hardware accelerators.

[2] The non-IID data quagmire of decentralized machine learning, Hsieh et al., 2019

---

### Official Review · Reviewer_MdFM · 2023-10-28

**Soundness:** 3 good
**Presentation:** 2 fair
**Contribution:** 2 fair
**Rating:** 5
**Confidence:** 3

**Summary:**

The authors propose BAFFLE, a backpropagation-free approach for federated learning. The key idea is to let each client perturb the model multiple times and conduct forward propagation for all models. Then, the server aggregates the loss differences to estimate the gradient. Experimental results demonstrates the effectiveness of BAFFLE in terms of memory efficiency, although it achieves lower performance than the full backpropagation.

**Strengths:**

1. The authors tackles an important problem, which is to avoid backpropagation in resource-limited federated learning clients.

2. The idea of the paper is easy to follow.

**Weaknesses:**

1. It is not clear what the theoretical results in Theorems 3.1 & 3.2 are exactly saying. The authors should provide more details on what is the theorem saying, and why it is guaranteeing the convergence of the algorithm. Moreover, are the authors assuming a strongly convex function or a non-convex loss function? Or does it holds for both?

2. In experiments, the authors show that BAFFLE is more resource efficient than full backpropagation, while achieving a lower accuracy, which is natural. Now the question is, in which environments should we use BAFFLE, and when is full backpropagation better? There is no clear guidance on this. One thing the authors can investigate is the accuracy vs. resource (e.g., client-side computation or overall delay, etc) plot. Depending on the system parameters, it would be meaningful if the authors can provide when BAFFLE should be utilized instead of full backprop.

3. Does BAFFLE also works with partial client participation?

4. I'm currently not sure why evaluate the scheme in a federated learning setup. Can this idea be applied in a centralized setup where all data samples are gathered in a single user? What is the performance under this setup?

**Questions:**

Please see weakness above.

---

### Official Review · Reviewer_vF9m · 2023-10-31

**Soundness:** 2 fair
**Presentation:** 2 fair
**Contribution:** 2 fair
**Rating:** 3
**Confidence:** 4

**Summary:**

This paper introduces a federated learning method called BAFFLE which replaces backpropagation by multiple forward or inference processes to estimate gradients. In this framework, each client downloads random seeds and global parameter updates from the server, generates perturbations locally based on the seeds, executes multiple forward propagations or inferences to compute loss differences, and then uploads the loss differences back to the server. Besides, since BAFFLE only utilizes forward propagations, which are memory efficient and do not require auto-differentiation, trusted execution environments (TEEs) can be used in this model to prevent white-box evasion. In the experiments, BAFFLE is used to train deep models from scratch or to finetune pre-trained models. Compared with conventional Federated Learning, the performance is acceptable.

**Strengths:**

Using zeroth-order optimization, this model replaces backpropagation with multiple forward or inference processes to obtain a stochastic estimation of gradients. Since this model is backpropagation-free, the communication costs and the computational and storage overhead for clients can be reduced. Because of this, TEEs which are highly memory-constrained can also be applied.

**Weaknesses:**

Some notations are not well defined when using them. The organization of this paper can be improved. BP baselines are not very clear in each experiment (FedAvg or FedSDG). The tested models are not large. The comparison between BAFFLE and state-of-the-art algorithms is missing. In PRELIMINARIES Zeroth-order FL, some numbers are provided to compare BAFFLE with FedZO, but in EXPERIMENT the comparison to other related works is not well presented. The robustness experiments are not convincing enough (not quantitative) and there is no comparison between BAFFLE and conventional FL.

**Questions:**

1. Have you considered BAFFEL's performance when the models are more complicated and have more parameters?

2. What if the number of clients is much larger?

---

### Official Review · Reviewer_7JpN · 2023-11-02

**Soundness:** 2 fair
**Presentation:** 3 good
**Contribution:** 2 fair
**Rating:** 3
**Confidence:** 4

**Summary:**

This paper proposes BAFFLE, a zeroth-order federated learning method with secure aggregation and TEE. BAFFLE benefits from the BP-free local optimization process and thus is more memory-efficient than traditional BP-based algorithms. Empirical studies demonstrate the effectiveness and the low memory costs of the proposed BAFFLE.

**Strengths:**

1. The paper is well-structured, and the methodology part is easy to follow.
2. The experiment section contains completed ablation studies for understanding the proposed method.

**Weaknesses:**

1. The overall contribution is limited. Zeroth-order optimization in Federated Learning has been already studied, and applying secure aggregation is straightforward as well. The novelty of theoretical analysis also seems limited. It would be better for the authors to summarize the comparison and improvements in terms of the algorithm side with FedZO (Fang et al., 2022).

2. The motivation for applying TEE in the proposed method is unclear. The paper only mentioned that TEE is memory-constraint and back-propagation is memory-consuming as well. However, it cannot be concluded that TEE can benefit from the BP-free method. I would like to see a more detailed explanation or just an applicable explanation about utilizing TEE in the BAFFLE algorithm instead of adding it as an extension.

3. Compared to other FL works, the experiments lack some settings. It seems that experiments for both i.i.d. and non-i.i.d. are conducted under client full participation settings, which is more closely to the centralized settings.

**Questions:**

1. What is the convergence analysis stating? Theorem 1 and 2 show that the zeroth-order gradient obtains an unbiased estimation for the true gradients with a convergence rate. However these results seem common in (centralized) zeroth-order optimizations. How do the theorems contribute to the proposed method?

2. Can you report the comparison between real-world computation costs (such as wall-clock time) for zeroth-order and BP algorithms, especially when K is large, Is the proposed method really time-consuming? Ideally, as presented in the paper “BAFFLE results in approximately K/5 times the computation expense of BP-based FL”, but it would be convincing if there is experimental evidence. Additionally, what is the number of local steps for your BP baseline (FedAvg or FedSGD)? It is crucial to point this number of local steps out. A simple example: Suppose “BAFFLE results in approximately K/5 times the computation expense of BP-based FL” holds correctly. If conducting K=5000 steps of BAFFLE for 20 epochs can achieve comparable accuracy as conducting 100 steps of FedAvg for 20 epochs. BAFFLE still takes 10 times more computational costs than FedAvg.

3. The writing related to experiments should be more clear to avoid misleading. In Section 2, “...95.17% accuracy on MNIST with 20 communication rounds versus 83.58% for FedZO with 1,000 rounds. “ How much is the K value in this comparison for BAFFLE? According to Section 4, “Specifically, we use Adam to train a random initialized model…”, what this Adam optimizer uses for? Does the optimizer for local training keep the same for BAFFLE and BP-baseline?

---

### Meta-Review · Area_Chair_iqUW · 2023-12-17

**Metareview:**

Summary: this paper studies 0th order optimization in the federated setting. Their algorithm involves each client evaluating the loss at K perturbations of the current parameter set to estimate the client's version of the gradient, and then using this in a way similar to regular (gradient based) FL in terms of aggregation.

Strengths: clear presentation of the algorithm

Weaknesses: The algorithm is not very novel, the analysis is similar to centralized case of showing that per-client gradient estimates come close to what they would be if we did gradients-based FL. small scale experiments where the (classical) gradient-based approach seems far from the known state of art for those networks and datasets.

**Justification For Why Not Higher Score:**

See summary weaknesses above, and also the many pointed out by the reviewers (who also gave it low scores across the board). No rebuttal or discussion from the authors to any of these issues.

**Justification For Why Not Lower Score:**

already has lowest score

---

### Decision · Program_Chairs · 2024-01-16

Reject